# Shrinkage and Durability of Waste Brick and Recycled Concrete Aggregate Stabilized by Cement and Fly Ash

**DOI:** 10.3390/ma15103684

**Published:** 2022-05-20

**Authors:** Yongfa Ding, Hongbo Li, Hubiao Zhang, Sheng Li, Xuanshuo Zhang, Shudong Hua, Jing Zhao, Yufei Tong

**Affiliations:** 1College of Civil and Hydraulic Engineering, Ningxia University, Yinchuan 750021, China; nxumrding@163.com (Y.D.); zhanghubiao1222@163.com (H.Z.); 13995891649@sina.cn (S.L.); zxsnikea@163.com (X.Z.); z17341244772@163.com (J.Z.); tongyufei1028@163.com (Y.T.); 2Engineering Research Center for Efficient Utilization of Water Resources in Modern Agriculture in Arid Regions, Yinchuan 750021, China; 3Ningxia Research Center of Technology on Water-Saving Irrigation and Water Resources Regulation, Yinchuan 750021, China; 4Ningxia Huasheng Energy Saving and Environmental Protection Technology Co., Ltd., Yinchuan 750021, China; ytmn0420@gmail.com

**Keywords:** regeneration gravel, shrinkage test, durability, microscopic analysis, ultrasonic test, ratio of Ca to Si

## Abstract

To study the shrinkage and freeze-thaw durability of cement-fly-ash-stabilized brick and concrete reclaimed gravel mixture (CFRBCA), recycled gravel was used to replace 100% of the natural gravel in cement-and-fly-ash-stabilized gravel (CFRCA). Five different mixture ratios of recycled brick and recycled concrete were designed. Dry shrinkage tests, temperature shrinkage tests, freeze-thaw cycle tests, ultrasonic tests, and microscopic analyses were then conducted. The test results showed that the water loss rate, dry shrinkage strain, and dry shrinkage coefficient of CFRBCA increased as the age and brick content increased and tended to be stable by approximately the 40th day of age. The reclaimed gravel content had a great influence on the temperature shrinkage of CFRBCA: the temperature shrinkage coefficient first increased and then decreased as the temperature decreased and reached a peak at −10 to 0 °C. The microstructure analysis showed that as the number of freeze-thaw cycles increases, cracks appear and extend in the CFRBCA, hydration products gradually change from dense to loose, and the Ca/Si ratio increases. Through these experiments, the logarithmic relationship model between ultrasonic wave velocity and CFRBCA strength damage, which can better predict the strength loss caused by CFRBCA variation with freeze-thaw cycles, was established. The brick content is the key parameter affecting the durability of the freeze-thawed CFRBCA, and thus the brick slag content should be properly controlled in engineering applications.

## 1. Introduction

The recycling of industrial and construction solid waste for road engineering has contributed to alleviating the global energy crisis and mitigating the effects of climate change. As a basic building material, cement-and-fly-ash stabilized gravel (CFRCA) has been used in many construction applications, such as road bases, foundation reinforcement, and dam construction [1,2,3,4]. Studies have shown that CFRCA has a good carrying capacity and that if used as a road base, it has great potential in improving the structural carrying capacity of an asphalt pavement [5,6,7,8]. However, CFRCA is easily affected by environmental factors and is prone to shrinkage cracks in areas that undergo large differences in temperature and large differences in dry and wet conditions [9,10,11]. The durability of CFRCA is tested when it is applied in seasonally frozen soil regions [12,13]. It is, therefore, of great importance to study the shrinkage and durability of CFRCA to optimize the performance of its mixtures.

Several research studies have already been conducted on cement-fly-ash-stabilized macadams. For example, Li et al. [14] examined the fractal characteristics, cumulative strain, and compressive strength of two kinds of reclaimed gravel under load and determined the practical significance of using a mixture of brick and concrete gravel with proper gradation. They also inferred that the dry shrinkage coefficient and temperature shrinkage coefficient of the material are key parameters for controlling crack generation during the service of the mixture and that it is very important to study these parameters for every new type of mixture applied to road bases. Xuan et al. [15] studied the effects of masonry content, cement content, water content and compaction degree on the shrinkage of the materials treated with abstract cement, and the results showed that stone content was the main factor affecting the dry shrinkage and thermal expansion performance of cement stabilized recycled brick structures. Peng et al. [16] used cement, fly ash, lime, and other inorganic materials, such as binders of recycled asphalt pavement materials, and conducted fatigue tests on three additives used to stabilize the recycled mixture. Their results showed that the recycled mixture has good temperature shrinkage resistance and dry shrinkage resistance. Furthermore, the abilities of different recycled gravel mixtures to resist freeze-thaw cycles differ in the base, and thus the freezing resistances of construction mixtures need to be comprehensively evaluated based on the macroscopic surface phase, microscopic damage, and mechanical indexes [17,18,19,20,21,22]. Trottier et al. [23] studied the freeze-thaw durability of recycled concrete aggregate using the direct displacement method, equivalent volume method, and particle accumulation model. Tang et al. [24] analyzed the influence of freeze-thaw cycles on the internal pore structures of soil-rock mixtures via nuclear magnetic resonance (NMR), and established the relationship between pore structure and shear characteristics under the freezing-thawing mechanism: as the number of freeze-thaw cycles increases, the shear strength parameters of the soil-rock mixture first decrease and then increase and finally decrease, whereas the porosity first increases and then decreases, and the material exhibits fractal characteristics. Ding et al. [25] proposed a six-parameter expression that accounts for the influence of pulse angle and hydrostatic pressure to solve the existing theoretical deficiencies in concrete damage specific strength and proved in their tests that the modified concrete criterion is better able to fulfill the corresponding characteristics of the failure surface. In addition, to solve the existing aging problem in the application of construction and industrial solid waste to road bases, a number of researchers have experimented with adding rubber powder, rubber fiber, and polymer materials into the mixture [26,27,28,29].

With the acceleration of China’s modernization, the annual output of construction waste has been increasing at an alarming rate. By the end of 2021, the total amount of construction waste in China was close to 5 billion tons, causing great pressure on the environment [30]. For example, in Ningxia, an important construction province in western China, there is presently a large amount of construction waste piled up in the open air because of annual immigration demolitions and residential renovations. These construction wastes occupy large amounts of land resources and are destroying the ecological environment. In line with the objective of recycling and utilizing construction waste, this study investigates the shrinkage and durability of cement–fly-ash-stabilized brick and concrete reclaimed gravel mixtures (CFRBCA) applied to road bases.

## 2. Raw Materials

### 2.1. Chemical Compositions

Fly ash was obtained from the Xixia District thermal power plant in Yinchuan, Ningxia. Brick and concrete recycled crushed stone were obtained from a local building centralized disposal center. The cement used is ordinary Portland cement P·O 42.5. The recycled brick and recycled concrete were crushed and screened using a crusher, and fine material with particle sizes less than 2.5 mm was obtained under the screen. The raw material, which was sampled via the quartering method, was tested for its chemical composition using an X-ray fluorescence analyzer. The test results are shown in Table 1. The main properties of cement are shown in Table 2 below.

Table 1 shows that SiO_2_, Al_2_O_3_, and Fe_2_O_3_ account for more than 75% of the chemical composition of fly ash by mass, which fulfills the technical requirements for the base and base binder. On the other hand, the combined SiO_2_ and Al_2_O_3_ content in the brick powder was more than 65%, indicating that the brick was a silicon-aluminum material with potential cementitious activity and pozzolan activity [31].

### 2.2. Basic Physical Indicators

Gravel was regenerated via the crushing and screening of the brick and concrete, and its basic physical properties were determined with respect to two divisions of particle size. The measurement results are shown in Table 3. The test data show that the brick rubble density was less than that of concrete regenerated gravel, whereas the bibulous rate of the former was greater than that of the latter. The reclaimed gravel should be saturated with water before the preparation of the mixture.

## 3. Test Scheme and Methods

### 3.1. Test Scheme Design

The cement content of cement and fly ash stabilized materials should be within the range of 3–7%; if the cement content is greater than 7%, it is not economically viable, whereas if the cement content is less than 3%, it would be difficult for the strength of the mixture to fulfill the requirements for road performance. The proportion of cement and fly ash should be within 1:3–1:5, whereas the best fly ash content is within 10–20% [32,33] Through preliminary adaptation, the cement content is determined to be 4%, whereas the fly ash content is determined to be 15%. The recycled gravel aggregate of brick and concrete is divided into four grades, in descending order of size: 26.5–31.5 mm, 16–26.6 mm, 4.75–16.0 mm, and 0–4.75 mm. The aggregate ratio of these grades is 5:24.5:35.5:35. The best water content and maximum dry density for five groups of CFRBCA were determined via a compaction test, the results of which are shown in Table 4.

### 3.2. Dry Shrinkage and Temperature Shrinkage Test Method

In accordance with the strain gauge method documented in the Test Specification of Inorganic Binder Stabilized Materials for Highway Engineering (JTGE51-2009), temperature shrinkage tests were conducted on the mixture in five proportions. Each sample was a Φ100 mm × 100 mm cylindrical specimen, with a compaction coefficient of 0.98. The mixture specimen was placed in a standard curing box with a temperature of 20 ± 2 °C and humidity of 95% for 6 d and soaked in water for 1 d, after which the surface moisture was dried and baked in an oven at 105 °C for 12 h to a constant weight. Before resistance strain gauges were pasted, sandpaper was used to smooth the pasting area. Vertical and annular strain gauges were then pasted onto the area, after which a lead wire was connected to the static strain gauge. The test temperature was controlled to be between 60 °C and −20 °C. Starting from a high temperature, the temperature was reduced step by step at a cooling rate of 1 °C/min to a temperature difference of 10 °C. The corresponding shrinkage of the specimen was then measured. Three parallel tests were conducted for each ratio, and the measurement results were averaged. The temperature shrinkage test is shown in Figure 1a.

In accordance with the drying shrinkage test method documented in the Test Specification of Inorganic Binder Stabilized Materials for Highway Engineering, dry shrinkage tests were conducted on the mixture in five ratios. The size of the specimen was the same as that used in the temperature shrinkage test. For each proportion, six specimens were prepared, three of which were used to measure the shrinkage deformation, whereas the remaining three were used to measure the dry shrinkage water absorption. After the specimens were formed, the standard curing test block was saturated with water for 7 d. The test block was then taken out, and the surface was air-dried until no evident traces of water were observed, to measure the initial mass and height of the specimen. A plexiglass sheet was then pasted, and a contrasometer was installed. Finally, the specimen and contrasometer were put together in a contraction chamber, and the micrometer was set to zero for data collection. Figure 1b presents the dry shrinkage test diagram of the mixture. From days 0 to 7, measurements were obtained once a day: the dial meter reading was recorded along with the weight of the test block. From days 7 to 30, the readings were obtained every two days. Subsequently, the data were collected on days 60, 90, 120, and 150.

### 3.3. Freeze-Thaw Cycle Test Method

In accordance with the freezing-thawing test method documented in the Test Specification of Inorganic Binder Stabilized Materials for Highway Engineering, freeze-thaw cycle tests were conducted on the mixture in five ratios. In accordance with the requirements set in the test regulations, 18 parallel specimens, including nine freeze-thaw specimens and nine standard-curing specimens, were prepared for each mixture ratio. After the preparation, the specimens were sealed in plastic bags and then placed in a standard curing box for 27 d. On day 27, the specimens were immersed in water for 24 h, after which the specimens were taken out for a compressive strength test. The compressive strength test adopts the WA-1000B series electrohydraulic servo universal testing machine, the maximum test force is 1000 kN, and the relative error of the value is ±1%. The specimens were then placed in the refrigerator for the freeze-thaw cycle test. The refrigerator temperature was set to −18 °C. For each freeze-thaw cycle, the specimens were frozen for 16 h and then thawed for 8 h in a 20 °C water bath. The freeze-thaw cycle tests were conducted for five mixture proportions, and the strength and mass changes of the specimens were recorded after 0, 5, 10, 15, 20, 25, 30, 35, and 40 freeze-thaw cycles. The compressive strengths were averaged to analyze the influencing factors for the frost resistance coefficient. The calculation methods for the frost resistance coefficient and mass loss rate are shown in Equations (1) and (2), respectively, as follows. The freeze-thaw cycle test is shown in Figure 2. Samples were taken from mixtures with different freezing-thawing cycles for SEM and EDS detection.
(1)$$BDR=\frac{R_{DC}}{R_{C}}\times 100$$

In the formula:

*BDR*—Frost resistance coefficient

*R_DC_*—Compressive strength of specimens after *n* cycles (MPa)

*Rc*—Compressive strength of control group specimens (MPa)
(2)$$W_{n}=\frac{m_{0}-m_{n}}{m_{0}}$$

*W_n_*—Change rate of specimen mass after *n* freeze-thaw cycles (%)

*m*_0_—Quality of specimen before freezing and thawing (g)

*m_n_*—Quality of specimen after *n* freeze-thaw cycles (g)

## 4. Test Results and Analysis

### 4.1. Temperature Shrinkage Test

Table 5 outlines the average vertical and annular shrinkage coefficients (TS) and maximum shrinkage coefficients (TS_max_) of the mixtures, whereas Figure 3 illustrates the vertical and annular temperature shrinkage coefficients.

According to the results of the temperature shrinkage test, the mixtures with the five different mixing ratios exhibited the same trend over the same temperature range, in that the temperature shrinkage coefficient first increased and then decreased as the temperature was decreased. Within the temperature range of 10–60 °C, the temperature shrinkage coefficient increased as the temperature decreased. The temperature shrinkage coefficient increased significantly within the temperature range of −10 – 10℃, and peaks were observed within the temperature range of −10–0 °C, indicating that the deformation of CFRBCA is most severe within the temperature range of −10–0 °C, where shrinkage cracks are prone to occur. Therefore, it is reasonable to evaluate the temperature shrinkage performance of CFRBCA for the temperature range of −10–0 °C. In addition, the temperature shrinkage coefficient increased as the brick ratio increased, with the most evident change occurring in the temperature range of 0–10 °C. As the brick ratio increased by 10%, the vertical temperature shrinkage coefficient of CFRBCA increased by 12.3%, 19.5%, 15.3% and 14.2%, respectively, whereas the circumferential temperature shrinkage coefficient increased by 12.8%, 9.4%, 41.4%, and 3.7%, respectively. It can be observed that as the brick content was increased from 30% to 40%, the circumferential temperature shrinkage coefficient changed significantly, and CFRBCA became prone to producing vertical cracks. Based on the findings thus far, a brick content of 30% can be used for evaluating and controlling the shrinkage deformation of CFRBCA.

### 4.2. Dry Shrinkage Test

Drying shrinkage is an internal phenomenon that the inorganic binder undergoes and is caused by moisture content change, which is manifested as volume shrinkage. In a cement stabilized material, water can exist in many forms, such as liquid water, interlayer structure surface adsorbed water, capillary water, and water in the process of curing caused by evaporation, capillary tension, and adsorption forces between the water and the macroscopic volume change as a whole. In this experiment, the relationship between drying shrinkage deformation and time for the CFRBCA was explored through dry shrinkage tests on the mixture in five ratios. Figure 4 shows the relationship between drying shrinkage strain, drying shrinkage coefficient, and water loss rate time.

The water in the cement-stabilized base exists in different forms, such as free water and bound water. When the specimen is exposed to air, the water will evaporate gradually, causing the water content of the material to decrease continuously. According to Figure 4a, the curve of the relationship between curing time and the water loss rate is as follows: a rapid growth rate in the early stage, a slow growth rate in the middle stage, and a gentle growth rate in the late stage. The specimen had the fastest growth rate 7 days before the test, reaching more than 60% of the total water loss rate, after which the growth rate of the water loss gradually slowed down from day 7 to day 40, and finally leveled off from day 40 to day 150. This is mainly because, in the early stage of the test, free water inside the specimen evaporated quickly, resulting in fast water loss. As the curing age increased, the specimen was exposed to the air for a long time, and the hydration reaction in the mixture consumed part of the free water, leading to a reduction in the free water evaporation of the mixture. Part of the free water was thus converted to bound water, and thus the water loss rate gradually decreased. In the later stage of the test, the free water and bound water had been exhausted, and thus the water loss rate of the mixture tended to be flat. As the brick content was increased, the water loss rate of the mixture also increased, which was mainly because the water absorption rate of the brick aggregate was much higher than that of the concrete reclaimed gravel. Based on these findings, the order of water loss rate for CFRBCA is as follows: CFRBCA5 > CFRBCA4 > CFRBCA3 > CFRBCA2 > CFRBCA1.

The dry-shrinkage strain and dry-shrinkage coefficient are used to characterize the sensitivity of a material to water: the larger the dry-shrinkage strain and dry-shrinkage coefficient of a material, the greater its sensitivity to water, indicating that the material is more likely to crack in the process of continuous water loss. It can be observed in Figure 4b,c that the drying shrinkage strain of 7d CFRBCA increased the fastest before the test, and both the drying shrinkage strain and drying-shrinkage coefficient reached more than 60% of the overall level. Subsequently, the growth gradually slowed down from day 7 to day 40, after which the drying shrinkage strain and drying shrinkage coefficient hardly increased. As the brick content was increased, the dry shrinkage strain and dry shrinkage coefficient increased, which was mainly because the dry shrinkage strains and dry shrinkage coefficients of cement and fly ash stabilized materials are closely related to the initial water content of the mixture. Increasing the brick ratio improves the optimal moisture content of the mixture, leading to increases in the dry shrinkage strain and the dry shrinkage coefficient. Therefore, the reclaimed brick gravel content in CFRBCA should not be excessively large. According to the relation diagram between the dry shrinkage coefficient and the water loss rate of the mixture, the dry shrinkage coefficient increases as the cumulative water loss rate increases. The hydration and evaporation of some water in the cement stabilized gravel base mixture reduced the moisture in the mixture, increased the capillary tension in the mixture, increased the activity of the adsorbed water and intermolecular force; the interlayer water action of mineral crystals or gel and carbonization dehydration caused the macroscopic volume change and the drying and shrinkage of the mixture. Thus, in practical engineering applications, CFRBCA should be watered for the first 40 days to reduce shrinkage cracks caused by drying and water loss.

### 4.3. Freeze-Thaw Cycle Test

#### 4.3.1. Influence of Freeze-Thaw Cycle on Microstructure

The CFRBCA3 test blocks were selected for further examination of the effects of the freeze-thaw cycle, and fine aggregate from inside the test blocks was sampled for microscopic detection and analysis. Axia ChemiSEM was used for scanning electron microscopy, the magnification was 5 to 100,000 times, the X-ray working distance was 10 mm, and the EDS detection Angle was 0 to 35°. Scanning electron microscopy (SEM) images of the mixture for different freeze-thaw cycles are shown in Figure 5.

Figure 5a–e shows the microscopic morphology of CFRBCA3 after different freeze-thaw cycles. As can be observed from the SEM images, CFRBCA3 was composed mainly of cementing materials, cracks, pores, and residual undissolved fly ash particles. The different forms of these cementitious substances and fly ash particles interwove and connected to form the whole. As shown in Figure 5a, the surface of the specimen mortar that did not undergo freezing and thawing was smooth, without evident cracks and holes, and the hydration products and internal microscopic morphology features were clearly observable. By contrast, after a number of freeze-thaw cycles, the CFRBCA3 samples were damaged to varying degrees, with cracks and pores generated in the samples. Furthermore, as the number of freeze-thaw cycles increased, these cracks extended and expanded, resulting in declines in the bonding performance and compactness of the internal structure. After 10 freeze-thaw cycles, microcracks appeared in the interface transition zone bonded by CFRBCA3. After 20 freeze-thaw cycles, the micro-cracks gradually extended and widened, and the fracture characteristics became evident. After 30 freeze-thaw cycles, the number of cracks in the interface transition zone increased, and after 40 freeze-thaw cycles, the cracks became wider and longer, and holes appeared in the interface transition zone. Analysis shows that, as the number of freeze-thaw cycles increases, water molecules enter the internal void of the specimen, and swelling occurs as the material freezes. Subsequently, fine aggregate falls off at the interface between the reclaimed aggregate and cement mortar, thus forming cracks [34]. Furthermore, the hydration reaction of CFRBCA3 generates part of the pin-rod ettringite (AFt). During the freezing-thawing process, water absorption expansion and water loss contraction occur in the AFt, easily resulting in volume expansion stress. When the expansion stress is greater than the bonding force between the reclaimed aggregate and cement mortar in the interface transition zone, cracks will occur. In addition, depending on the number of freeze-thaw cycles, the mixture sample will exhibit different degrees of alkali spreading, and the soluble salt and alkali in the test block will be precipitated in the capillary, resulting in crystallization pressure and promoting the cracking of concrete.

#### 4.3.2. EDS Spectrum Analysis

Energy-dispersive X-ray spectroscopy (EDS) is a method used to analyze the types and amounts of elements in the microregions of a sample. EDS analysis uses the same instrument as SEM. Freeze-thaw cycle tests were conducted on a CFRBCA3 test block after curing for 28 days under standard conditions. Gable substances in the cracks of mixtures subjected to different numbers of freeze-thaw cycles were selected for elemental analysis. The energy spectrum curves are shown in Figure 6a–d.

According to the results of the analysis, as visualized in Figure 6a–d, the elements O, Si, Al, Ca, C, S, Mg, Na, and Fe were present in the interface transition zone, among which elements O, Si, Al, and Ca were evenly distributed. Moreover, columnar products were observed in the interface transition zone of the mixture, and, according to EDS and SEM analyses, these hydration products may be C-S-H and Aft [35]. The Ca/Si ratios of hydration products after 0–30 freeze-thaw cycles were 0.94, 1.09, 1.05, and 1.35, respectively. After freezing and thawing, the Ca/Si ratios of the samples tended to increase with respect to those of the samples that were not subjected to freezing and thawing. Consequently, as the Ca/Si ratio increased, the hydration products changed from dense granular particles to loose granular particles, and the particle surface disintegrated into needle-rod shapes. The needle-shaped cylindrical AFt from among the hydration products of cement and fly ash, particularly because of the swelling properties of the AFt itself, can block-fill the mixture subjected to the frost heaving force produced under the action of cracks and pores, thus inhibiting further crack development, preventing the generation of cracks and pores, and stabilizing the compressive strength and macro performance of the connectivity within the mixture subjected to repeated freezing and thawing. The needle-column hydration products also included C-S-H and C-S-H with a lower Ca/Si ratio exhibiting a lamellar structure [36]. As the number of freeze-thaw cycles is increased, the calcium-silicon ratio increases and the lamellar structure disintegrates into needle-rod shapes.

#### 4.3.3. Influence of Freeze-Thaw Cycles on Strength

Unconfined compressive strength tests were conducted on the mixture specimens after the freeze-thaw cycles. Figure 7 visualizes the variation rule for the compressive strength of the mixture for different mix ratios with respect to the number of freeze-thaw cycles.

It can be observed from Figure 7 that the compressive strength of CFRBCA decreased as the number of freeze-thaw cycles increased. The compressive strength and number of freeze-thaw cycles conformed to a fitted function, with a correlation coefficient R^2^ greater than 0.97, reflecting a good correlation between the number of freeze-thaw cycles and unconfined compressive strength. The compressive strength of the CFRBCA with different compositions decreased as the brick content was increased. Specifically, the compressive strength of CFRBCA1 decreased linearly as the number of freeze-thaw cycles was increased, whereas the compressive strengths of CFRBCA2-CFRBCA5 decreased in a quadratic parabolic shape as the number of freeze-thaw cycles was increased. When the number of freeze-thaw cycles was between 0 and 20 times, the compressive strength of CFRBCA5 decreased as the number of cycles was increased. Furthermore, the strength of the material decreased significantly. By contrast, when the number of freeze-thaw cycles was greater than 20, the strength of the material decreased gently. This analysis showed that the porosity and water absorption of the brick aggregate were much higher than those of the concrete reclaimed gravel. The higher the brick ratio of the mixture, the greater the water absorption. The water in the pores of the mixture freezes and causes expansion stress, and the stress on the inner walls of the pores squeezes the material. As the number of freeze-thaw cycles increases, this stress gradually increases. An analysis of the macroscopic mechanical properties showed that the strength loss of CFRBCA becomes more significant as the number of freeze-thaw cycles is increased. On the other hand, after the number of freeze-thaw cycles reaches approximately 20, the change in compressive stress with respect to the number of cycles tends to be gentle, whereas the compressive strength tends to be stable.

Subsequently, a quantitative analysis of the freeze-thaw damage of CFRBCA was conducted. The variation trends in the frost resistance coefficient and the changes in quality for the five mixture ratios with respect to the number of freeze-thaw cycles are shown in Figure 8.

After the freeze-thaw cycles, the specimen surfaces exhibited evident damage. As the number of freeze-thaw cycles was increased, the mass loss of the mixture specimen first increased and then decreased in a parabolic shape. When the number of freezing and thawing cycles of CFRBCA1 and CFRBCA2 reached 30, the mass loss reached its peak, and the maximum mass loss rate was 4.5%. On the other hand, when the number of freeze-thaw cycles of CFRBCA3, CFRBCA4, and CFRBCA5 reached approximately 15, the mass loss reached its peak, and the maximum mass loss rate was 3.7%. As the brick-concrete ratio was increased, the number of freeze-thaw cycles required to reach the peak mass loss rate decreased.

The freezing resistance coefficient is the ratio of the compressive strength of a specimen before and after freezing and thawing and is one of the indexes used to measure the freezing resistance of a mixture. As shown in Figure 9, the frost resistance coefficient of CFRBCA decreased as the number of freeze-thaw cycles increased from 0 to 20. When the number of freeze-thaw cycles became greater than 20, the frost resistance coefficient decreased slowly as the number of cycles increased. As the brick ratio was increased, the frost resistance of CFRBCA became worse, which was mainly because the water absorption rate largely determines the frost resistance. The larger the brick ratio, the greater the water absorption rate of the specimen, and the worse the frost resistance of CFRBCA. After 20 freeze-thaw cycles, the pore walls of the specimen were damaged by frost heave, the trend of pore expansion decreased, and the pore size basically remained constant. Secondary hydration reactions inside the specimen began to strengthen the material as hydration products filled the pores, which enhanced the compressive strength of CFRBCA in the late freeze-thaw cycle and slowed the decreasing trend of the frost resistance coefficient. When the number of freeze-thaw cycles reached 40, the strength losses of the CFRBCA1–CFRBCA5 specimens reached 40.7%, 35.2%, 43.1%, 48.3%, and 51.3%, respectively.

#### 4.3.4. Ultrasonic Intensity Damage Detection

Studies have shown that the propagation speed of ultrasonic waves in a medium is closely related to the strength of the material. The higher the strength of the medium, the higher the wave speed of an ultrasonic wave passing through the medium; conversely, the lower the wave speed, the lower the strength of the medium. Establishing a relationship model between ultrasonic wave speed and strength damage due to freeze-thaw cycles can help to effectively predict the service performance of the mixture [37,38]. First, the unconfined compressive strengths of the mixture specimens for five different mixture ratios after different numbers of freeze-thaw cycles were measured. Afterward, the solid strength and ultrasonic wave velocity in the process of freezing and thawing were fitted using nonlinear function fitting, as shown in Figure 9. As shown in the figure, the strength losses of the mixture specimens subjected to different numbers of freeze-thaw cycles were negatively correlated with ultrasonic wave velocity; the larger the ultrasonic wave velocity, the greater the strength loss. During a freeze-thaw cycle, the strength damage of a mixture specimen is in a logarithmic relationship with ultrasonic wave velocity, with the fitting curve formula shown in Formula (3).
(3)y=ln(a−bx) (a, b are constants)

The correlation coefficients of the model fitting were all higher than 0.94, and the fitting effect was significant, which can better reflect the relationship between the amount of strength damage of a mixture specimen and ultrasonic wave velocity and can provide theoretical guidance for practical engineering applications.

## 5. Engineering Application

The experimental study aims to provide an effective application scheme for the comprehensive utilization of construction waste and industrial solid waste in Ningxia, China. In June 2020, Ningxia Huasheng Energy Conservation and Environmental Protection Technology Co., LTD., based on the research results of this experimental scheme, classified the collected construction waste, crushed and screened it for treatment, and prepared CFRBCA2 on site, which was, respectively, applied to the Silk Road water supply and drainage engineering of the industrial park in eastern Ningxia and the paving of the west Shenyang Road in Yinchuan, Ningxia. The recycling and utilization of construction waste are shown in Figure 10.

To test the practical engineering application effect of CFRBCA, strength tests were performed on the core sample of the field spreading base in March 2022. It was determined that the compressive strength was 10.3 MPa, the splitting strength was 0.74 MPa, and the average test result of the road bending settlement value was 0.32 mm. The Ningxia Yinchuan project has been active for two years, and yet the road surface has not exhibited shrinkage cracks, cracks, and other quality problems. Thus, this project is a relatively successful construction-waste application case. The most important technical problem with CFRBCA paving of road bases is not with strength losses and shrinkage cracks produced during seasonal climate changes; rather, it is with the moisture content of the mixture subjected to freeze-thaw damage. The Ningxia Yinchuan project is located in a temperate zone with a continental semi-arid climate and annual rainfall of less than 500 mm. Thus, it is a relatively appropriate CFRBCA application demonstration area.

## 6. Conclusions

Through experimental research on the shrinkage and durability of waste brick and recycled concrete aggregate stabilized by cement and fly ash, the following conclusions are drawn.

(1)As the temperature decreases, the temperature shrinkage coefficient of CFRBCA first increases and then decreases. The temperature shrinkage coefficient increases significantly in the temperature range of −10 to 10 °C and reaches a peak value in the temperature range of −10 to 0 °C. It is, therefore, reasonable to use a temperature shrinkage coefficient of −10 to 0 °C to evaluate the temperature shrinkage performance of CFRBCA. The temperature shrinkage coefficient of the mixture increases as the brick-concrete ratio is increased, where the change is most evident in the temperature range of 0 °C to 10 °C. When the brick-concrete ratio is increased from 30% to 40%, the temperature shrinkage coefficient of the mixture increases by 41.4% in the circumferential direction.(2)The water loss rate, dry shrinkage strain, and dry shrinkage coefficient of CFRBCA all increase with respect to time and brick-concrete ratio; the growth rate is the fastest in the first 7 days and tends to be stable at approximately 40 days of age. It is suggested that CFRBCA should be watered and maintained for 40 days before it is applied to a road base.(3)As the number of freeze-thaw cycles is increased, the mass loss of CFRBCA first increases and then decreases in a parabolic pattern, and the maximum mass loss rate is 4.5%. For 0–20 freeze-thaw cycles, the freezing resistance coefficient of CFRBCA decreases as the number of cycles is increased. Subsequently, after 20 freeze-thaw cycles, the freezing resistance coefficient decreases slowly as the number of cycles is increased. This is because the brick aggregate itself has a larger porosity and bibulous rate, and thus, it undergoes a larger strength loss during the freeze-thaw process.(4)The microscopic morphology of the mixture for different brick content was studied using a scanning electron microscope at different freeze-thaw cycles. The results showed that as the number of freeze-thaw cycles is increased, the microscopic morphology of the mixture exhibits evident change trends: the appearance of micro-cracks, increase in cracks, continuous extension of cracks, and pit corrosion. The main hydration products (C-S-H and AFt) in the crack areas of the mixture are also changed: the hydration products gradually disintegrate from dense granular shapes to needle-rod shapes. EDS analysis showed that the Ca/Si ratio of the hydration products gradually increased as the number of freeze-thaw cycles increased.(5)Through ultrasonic nondestructive testing, a compressive strength damage prediction model with respect to the freeze-thaw cycle was established. The strength damage of the mixed material was in a logarithmic relationship with ultrasonic wave velocity, and the fitting effect was significant, which can provide certain theoretical guidance in engineering practice. Therefore, the application of CFRBCA from Ningxia for road construction is proved to be feasible.

This paper systematically studied the shrinkage and freeze-thaw durability of the mixture, but the ion erosion resistance and erosion resistance of the material has not been studied. Therefore, corrosion resistance, erosion resistance and other tests related to the road performance of the material can be carried out in the next step.

## Figures and Tables

**Figure 1 materials-15-03684-f001:**
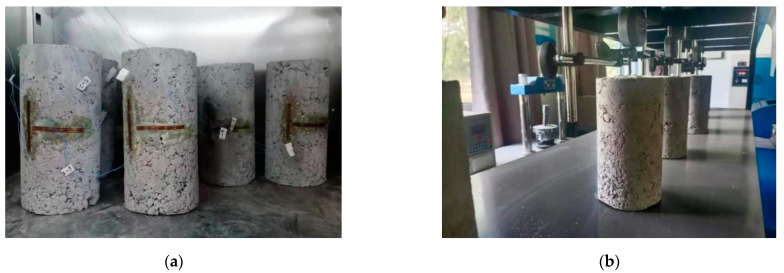
Shrinkage test: (**a**) Temperature shrinkage test, (**b**) Dry shrinkage test.

**Figure 2 materials-15-03684-f002:**
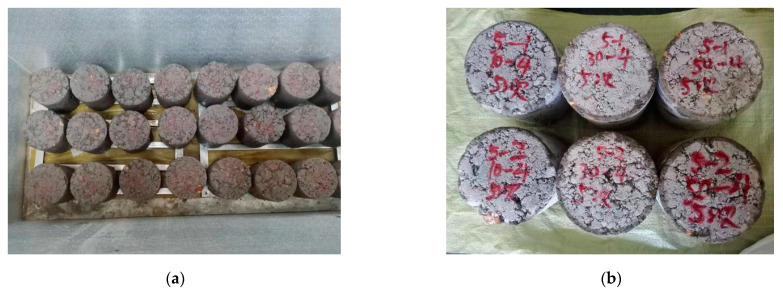
Freeze-thaw cycle test: (**a**,**b**).

**Figure 3 materials-15-03684-f003:**
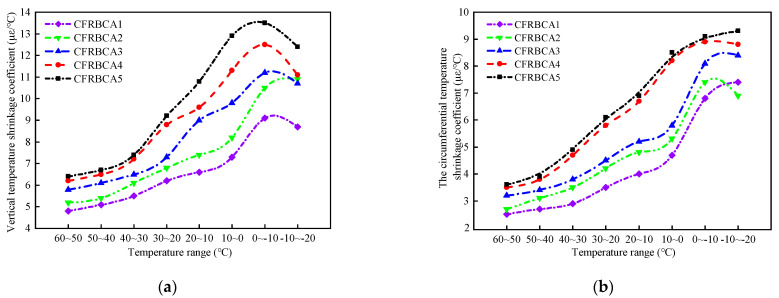
Relationship between temperature shrinkage coefficient and temperature range: (**a**) Vertical temperature shrinkage, (**b**) Circumferential temperature shrinkage.

**Figure 4 materials-15-03684-f004:**
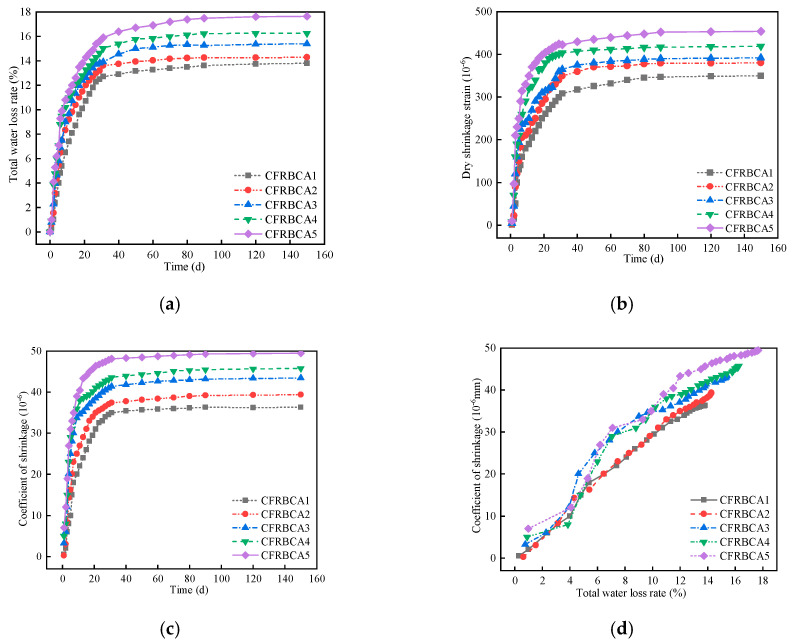
Dry shrinkage test results: (**a**) Relationship between total water loss rate and time, (**b**) Relationship between dry shrinkage strain and time, (**c**) Relationship between shrinkage coefficient and time, (**d**) Relationship curve between shrinkage coefficient and water loss rate.

**Figure 5 materials-15-03684-f005:**
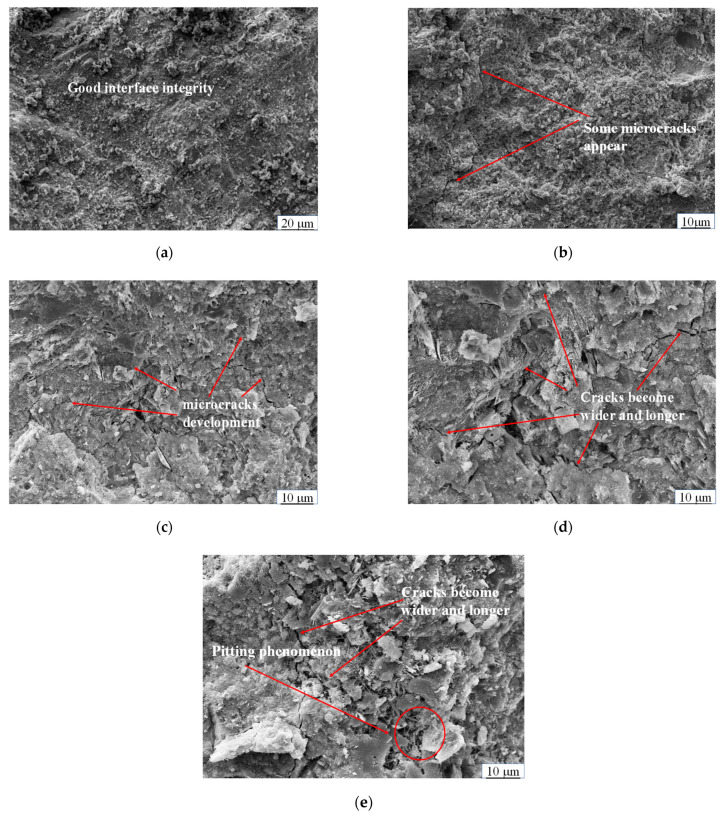
SEM images of CFRBCA3 at different freeze-thaw cycles: (**a**) 0 freeze-thaw cycles, (**b**) 10 freeze-thaw cycles, (**c**) 20 freeze-thaw cycles, (**d**) 30 freeze-thaw cycles, (**e**) 40 freeze-thaw cycles.

**Figure 6 materials-15-03684-f006:**
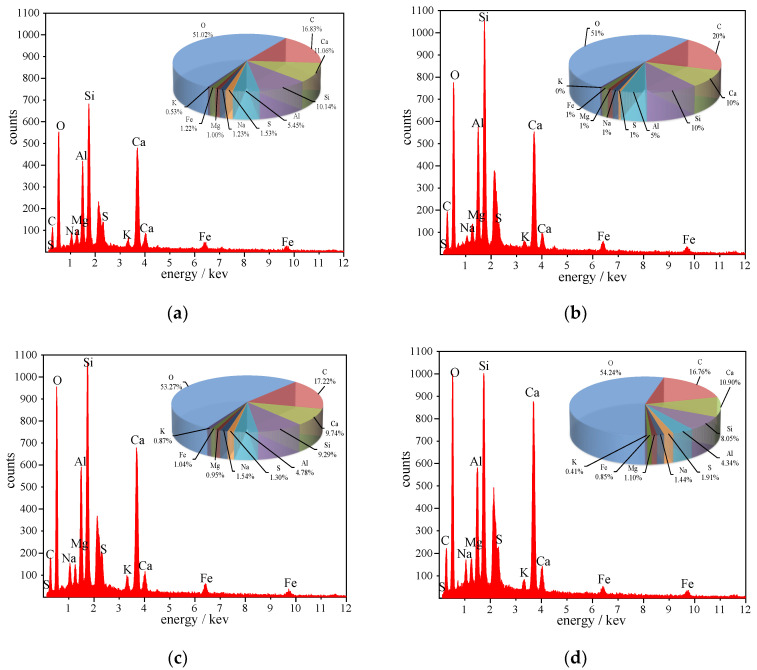
EDS spectrum analysis: (**a**) 10 freeze-thaw cycles, (**b**) 20 freeze-thaw cycles, (**c**) 30 freeze-thaw cycles, (**d**) 40 freeze-thaw cycles.

**Figure 7 materials-15-03684-f007:**
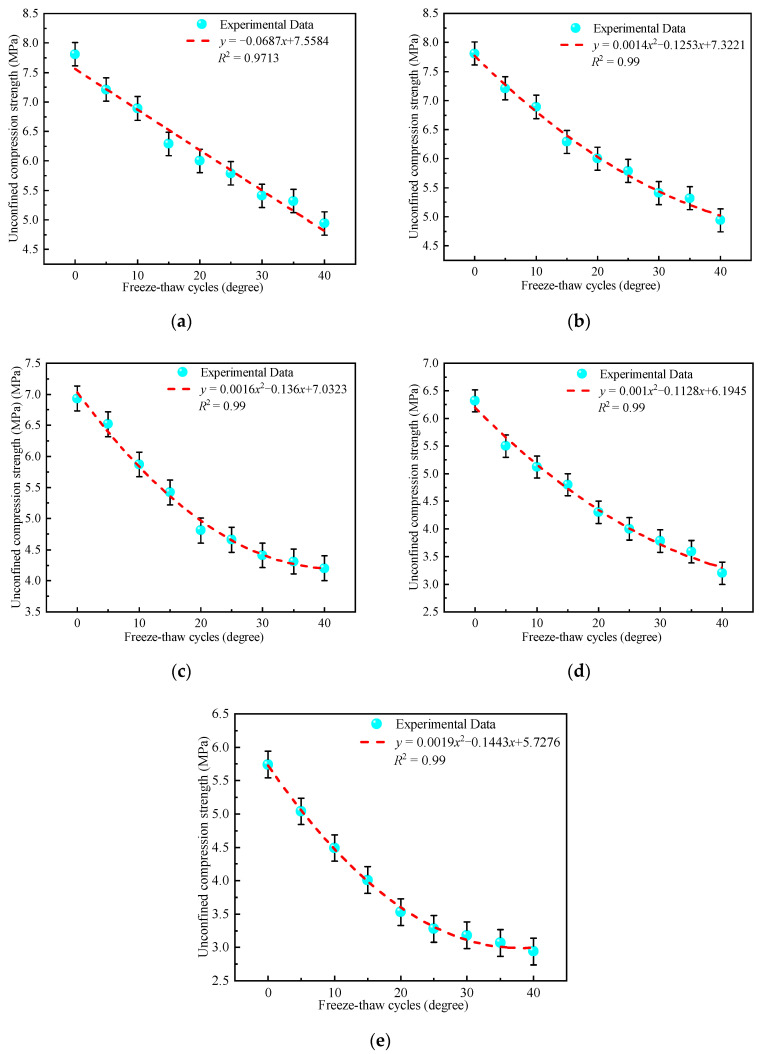
Relationship between unconfined compressive strength and freeze-thaw cycle: (**a**) CFRBCA1, (**b**) CFRBCA2, (**c**) CFRBCA3, (**d**) CFRBCA4, (**e**) CFRBCA5.

**Figure 8 materials-15-03684-f008:**
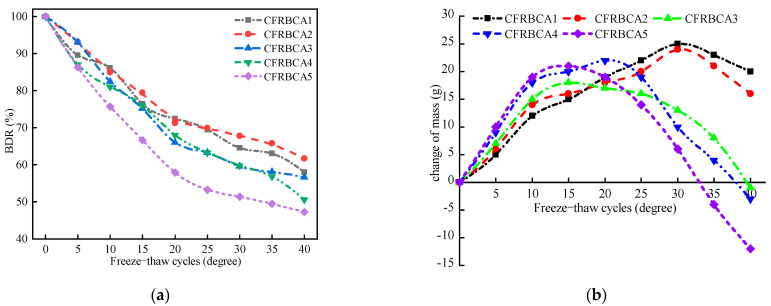
Relationship between freeze-thaw cycle with BDR and mass loss: (**a**) Freezing resistance coefficient of specimens for different freeze-thaw cycles, (**b**) Relationship between specimen mass and freeze-thaw cycle.

**Figure 9 materials-15-03684-f009:**
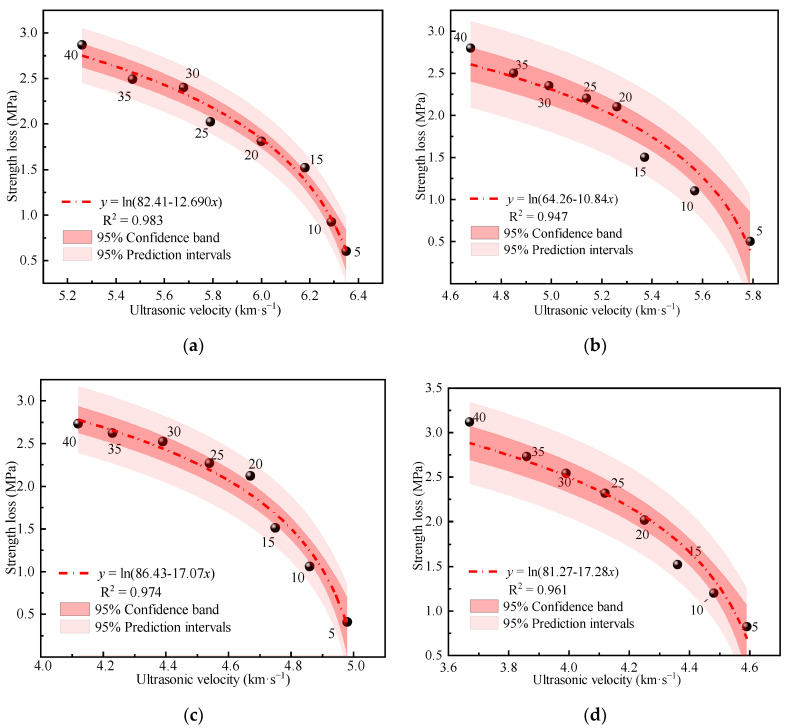
Relationship between unconfined compressive strength and freeze-thaw cycle: (**a**) CFRBCA1, (**b**) CFRBCA2, (**c**) CFRBCA3, (**d**) CFRBCA4, (**e**) CFRBCA5.

**Figure 10 materials-15-03684-f010:**
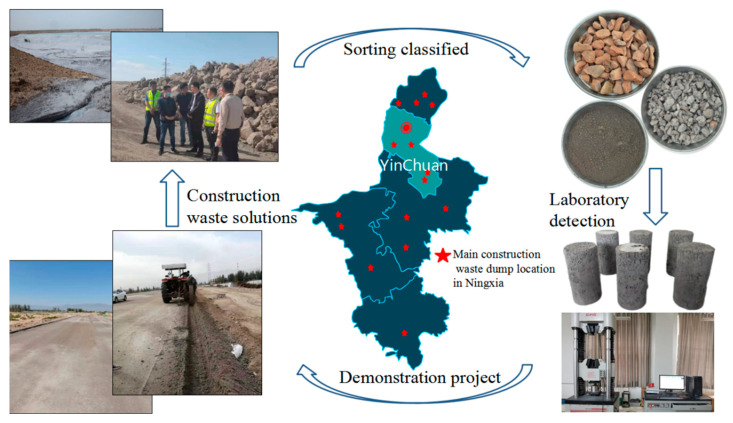
CFRBCA regeneration diagram.

**Table 1 materials-15-03684-t001:** Chemical compositions of raw materials (%).

Raw Materials	Mass Fraction								Ignition Loss
	SiO_2_	Al_2_O_3_	Fe_2_O_3_	CaO	MgO	K_2_O	Na_2_O	SO_3_	
Brick powder	35.81	37.66	4.40	7.63	3.41	2.94	2.69	0.78	6.7
Fly ash	28.56	37.92	12.8	12.28	2.10	1.69	1.48	0.54	3.4
Cement	22.36	6.83	3.81	60.74	1.25	1.60	0.59	0.67	3.1

**Table 2 materials-15-03684-t002:** Main performance index of cement.

Fineness Modulus/%	Setting Time/min		Compressive Strength/MPa	
	Initial setting time	Final setting time	3 d	28 d
1.6	225	375	25.48	42.67

**Table 3 materials-15-03684-t003:** Physical indexes of reclaimed gravel from brick and concrete.

Aggregate Category	Particle Size Range (mm)	Packing Density (g·cm^−3^)	Tapped Density (g·cm^−3^)	Apparent Density (g·cm^−3^)	Bibulous Rate (%)
Brick reclaimed gravel	<4.75	1.050	1.212	/	25.09
	4.75–26.5	0.767	0.881	2.35	23.87
Concrete reclaimed crushed stone	<4.75	1.468	1.650	/	5.94
	4.75–26.5	1.388	1.532	2.68	3.66

**Table 4 materials-15-03684-t004:** Physical indexes of reclaimed gravel for brick and concrete.

Test Number	Cement (%)	Fly Ash (%)	Ratio of Brick to Concrete	Optimum Water Content (%)	Maximum Dry Density (g·cm^−3^)
CFRBCA1	4	15	1:9	13.009	1.772
CFRBCA2			2:8	15.045	1.728
CFRBCA3			3:7	17.001	1.724
CFRBCA4			4:6	17.846	1.684
CFRBCA5			5:5	19.295	1.670

**Table 5 materials-15-03684-t005:** Average and maximum shrinkage coefficients.

Contraction Coefficient (µm/°C)		CFRBCA1	CFRBCA2	CFRBCA3	CFRBCA4	CFRBCA5
Vertical temperature shrinkage coefficient	TS	6.7	7.6	8.3	9.2	9.9
	TS_max_	9.1	10.9	11.2	12.5	13.5
Circumferential temperature shrinkage coefficient	TS	4.3	4.7	5.1	6.3	6.5
	TS_max_	7.4	7.4	8.1	8.9	9.3

## Data Availability

The data supporting the findings of this study are available from the corresponding author upon reasonable request.

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
