# Peer review of "Shrinkage and Durability of Waste Brick and Recycled Concrete Aggregate Stabilized by Cement and Fly Ash"

_materials, 2022, doi:10.3390/ma15103684_

Round 1

Reviewer 1 Report

Submitted manuscript outlined the application of demolition waste in the form of brick and concrete reclaimed gravel to the composite mixes stabilized with combined cement and fly ash binder suitable as a stabilizing and construction material of road bases. Together with given issue description, authors also provided the combination of reals experiments and numerical simulations helping to corelate the relationship between the strength loses of cylindric samples exhibited to freeze-thaw cycles loading and ultrasonic measurements.  Further, the temperature and draying shrinkage parameters were measured and discussed. The practical application of developed composites was showed as well. The manuscript is logically arranged, well written and readable. I have, however, few comments related to the quality of text pars as well as to missing description of applied experimental techniques/equipment, as follows:

Row 50; 56; 60; 67; 69; 75, in literature references abbreviations including more authors the dot is missing (e.g., Li et al.).

Row 97, the sentence containing information about used cement type should be rewrite, in order to improve its structure. In addition, some basic information about cement producer could be mentioned.

Table 1, is the chemical composition of fly ash right? It shows predominant content of silica dioxide exceeding 95 % and other components, in particular calcium oxide, are in minor amounts only.

Table 2, I would recommend to use the expression tapped density instead of tap density.

The ranges of values e.g., 3 – 7%, should be separated by gabs, not in the way appearing through text parts…3-7%. Revise it.

Row 122, the literature references are not separated by gab.

Row 134, use the proper symbol for diameter.

The unit of degrees of Celsius appearing through text parts is misinterpret. Revise it to the form of e.g., 20 °C.

Row 146, Rewrite the label of Figure one with the right usage of capital letters and comas. Current state is not acceptable.

Rows 185 – 187, abbreviations description contains two dashes. Erase one.   

In the part 3.3 compressive strength test are mentioned. What king of testing device was used? How were loading conditions? What was the standard deviation or measuring uncertainty of compressive strength tests, in view of real experiments?

Row 203, adjust temperature range notation, current form is confusing.

Part 4.2 and other text parts, I would recommend to use one term only and that is drying shrinkage.

Part 4.3.1, SEM measurements and corresponding data are mentioned. What king of testing device was used? What was its set up during measurements?

Part 4.3.2, X-ray measurements and corresponding data are mentioned. What king of testing device was used? What was its set up during measurements?

Row 405, the literature reference in not separated by gap from the relevant text part.

Row 415, the writing and expression of formula 3 make in the same way like in the case of formulas 1 and 2.

In conclusion part the first work introducing paragraph and final future motivation and outlook in the end part are missing.

Regard to mentioned above, given manuscript reaches high quality, and thus I can recommend it for the publication in the Materials journal, however, after minor revision, especially taking into account adding information of experimental procedures and correcting text parts.

Reviewer 2 Report

The topic of the research work and manuscript is really interesting and provides new information. However there are several issues to be addressed towards its quality improvement before thinking of publication. 

You should add dot in the end of each of "et al." used in the whole text. The phrase "ecological environment" in line 89 needs improvement. In my opinion, the state-of-the-art is not adequately described, while there are several previous studies dealing with cement and fly ash action, waste bricks, recycled materials, concrete in aggregates, road bases recycled materials etc. Most of the bibliography references used in the text are coming from a particular location of the planet (same country or neighbours to the authors), while there are several relevant international studies (Amirican, european etc.) that would be significant to be taken into consideration to be your approach more integrated and appropriate. I would also propose to the authors to incorporate in the theoretical approach significant information from the relevant and recent work DOI: 10.1016/B978-0-12-824543-9.00017-7, to support a proposed to be added brief statement on the presence of natural fibers in such aggregates. In the title and the end of introduction where you refer the word durability, please clarify if you are talking about the mechanical or biological durability (if it is mechanical durability you could clarify substituting to the word strength for example or explain). 

You should provide details concerning the crusher, X-ray fluorescence analyzer used, contrasometer, as well as for all the equipment used (model, manufacturer country). In line 99, do you mean 0 mm to 2.5 mm? Table 1 is providing results and should be transferred to results section. Which are the standards that the methodology was based on concerning these measurements that you describe? Concerning the chemical composition of the raw materials, did you made only one measurement? otherwise you should present the standard deviation as well in the tables. Please, provide more information concerning the workability/viscosity of the mixtures and the preparation of the samples (mixing process, setting duration, conditions etc. till the time of properties testing). In line 158, you rather use "presents" rather than "is". In line 160, the phrase "the quality of the specimen was weighed" needs improvement. The statistical analysis of the results is not described in the materials-methods chepter.

Reviewer 3 Report

In this study, the authors investigated the shrinkage and durability of waste brick and recycled concrete aggregate stabilized by cement and fly ash. Generally speaking, it is an interesting research with good information. However, there are some problems which the authors should look into.

  1. It should be "recycled concrete aggregate", maybe not "concrete recycled aggregate".
  2. In  the abstract, the description of test results should be done using the past tense.
  3. The recycling of industrial solid waste to make sustainable concrete should be given a more detailed review. In recent years, increasing interest has been paid to the alkali-activated materials (AAMs) or geopolymer because those sustainable binding materials are derived from waste materials. ("A state-of-the-art review of crushed urban waste glass used in OPC and AAMs (geopolymer): Progress and challenges. Cleaner Materials, p.100083.")
  4. The innovation and objective of this study should be emphasized. What is the background of using such a material?
  5. Why does the shrinkage of this material matter? How does it compare to the normal materials?
  6. More in-depth discussion should be provided on the shrinkage mechanism upon drying.

Round 2

Reviewer 2 Report

As I have checked the authors have implemented the proposed changes in the revised verion of manuscript towards the improvement of their work. Almost all the changes have been implemented and in my opinion, the manuscript is well-prepared and organized enough to be accepted for publication in this journal. I remain at your disposal for any clarification.

Reviewer 3 Report

This paper has been revised based on the comments and should be ready for publication.